# Gut Microbiota in Patients with Non-Alcoholic Fatty Liver Disease without Type 2 Diabetes: Stratified by Body Mass Index

**DOI:** 10.3390/ijms25031807

**Published:** 2024-02-02

**Authors:** Natthaya Chuaypen, Aisawan Asumpinawong, Pattarose Sawangsri, Jakkrit Khamjerm, Nutta Iadsee, Thananya Jinato, Sawannee Sutheeworapong, Suthep Udomsawaengsup, Pisit Tangkijvanich

**Affiliations:** 1Center of Excellence in Hepatitis and Liver Cancer, Department of Biochemistry, Faculty of Medicine, Chulalongkorn University, Bangkok 10330, Thailand; natthaya.ch56@gmail.com (N.C.); jakkrit.champer@gmail.com (J.K.); nutta.i@kkumail.com (N.I.); ji.thananya@gmail.com (T.J.); 2Metabolic Diseases in Gut and Urinary System Research Unit (MeDGURU), Department of Biochemistry, Faculty of Medicine, Chulalongkorn University, Bangkok 10330, Thailand; 3Treatment of Obesity and Metabolic Disease Research Unit, Department of Surgery, Faculty of Medicine, Chulalongkorn University, Bangkok 10330, Thailand; aisawan.asum@gmail.com (A.A.); spattarose@gmail.com (P.S.); suthep.u@chula.ac.th (S.U.); 4Biomedical Engineering Program, Faculty of Engineering, Chulalongkorn University, Bangkok 10330, Thailand; 5Medical Biochemistry Program, Faculty of Medicine, Chulalongkorn University, Bangkok 10330, Thailand; 6Systems Biology and Bioinformatics Research Unit, Pilot Plant Development and Training Institute, King Mongkut’s University of Technology Thonburi, Bangkok 10150, Thailand; s.sawannee@gmail.com

**Keywords:** gut microbiota, lean, obesity, NAFLD, diabetes

## Abstract

The relationship between gut dysbiosis and body mass index (BMI) in non-diabetic patients with non-alcoholic fatty liver disease (NAFLD) is not adequately characterized. This study aimed to assess gut microbiota’s signature in non-diabetic individuals with NAFLD stratified by BMI. The 16S ribosomal RNA sequencing was performed for gut microbiota composition in 100 patients with NAFLD and 16 healthy individuals. The differential abundance of bacterial composition between groups was analyzed using the DESeq2 method. The alpha diversity (Chao1, Shannon, and observed feature) and beta diversity of gut microbiota significantly differed between patients with NAFLD and healthy controls. However, significant differences in their diversities were not observed among subgroups of NAFLD. At the phylum level, there was no trend of an elevated *Firmicutes/Bacteroidetes* ratio according to BMI. At the genus level, patients with lean NAFLD displayed a significant enrichment of *Escherichia-Shigella* and the depletion of *Lachnospira* and *Subdoligranulum* compared to the non-lean subgroups. Combining these bacterial genera could discriminate lean from non-lean NAFLD with high diagnostic accuracy (AUC of 0.82). Non-diabetic patients with lean NAFLD had a significant difference in bacterial composition compared to non-lean individuals. Our results might provide evidence of gut microbiota signatures associated with the pathophysiology and potential targeting therapy in patients with lean NAFLD.

## 1. Introduction

Non-alcoholic fatty liver disease (NAFLD) is emerging as one of the most common chronic liver diseases, affecting approximately 25–30% of the global population [1]. NALFD is characterized by excess fat accumulation in the liver, with its manifestations ranging from simple steatosis to non-alcoholic steatohepatitis (NASH), leading to cirrhosis and hepatocellular carcinoma (HCC). Increasing evidence also demonstrates that NAFLD is a multisystem disease and generally coexists with metabolic disorders, including type 2 diabetes mellitus (T2DM), dyslipidemia, and insulin resistance. Although NAFLD was conventionally associated with obesity, a high proportion of individuals have normal weight according to their body mass index (BMI < 23 kg/m^2^ in Asians and <25 kg/m^2^ in non-Asians). This so-called ‘lean NAFLD’ is observed in 10–20% of Asian populations and is thought to have less metabolic dysfunction than obese individuals [2]. However, a recent systematic review and meta-analysis demonstrates that individuals with lean NAFLD have a significantly higher risk of liver-related complications compared to those with non-lean NAFLD [3]. These data indicate that further understanding of its pathogenesis is needed for individualized treatment strategies in this subgroup of NAFLD.

The mechanisms of NAFLD development and progression are multifactorial, including genetic susceptibility, metabolic disturbance, lifestyle, and environmental factors [4]. Additionally, recent data have suggested that gut microbiota plays an essential role in the pathophysiology of NAFLD. It has been shown that changes in gut microbial composition can influence and accelerate the severity of steatosis and fibrosis through the gut–liver axis [5]. Compared to healthy individuals, most data demonstrate that patients with NAFLD display distinct microbiota signatures with reduced bacterial diversity [6,7]. However, some discrepancies across reports are observed that might be attributed to the heterogeneity in studied populations, ethnicity, and geographical regions. Moreover, one of the challenges in studying the association between NAFLD and gut microbiota is the confounding effect of T2DM on gut dysbiosis since these two metabolic diseases are usually coexisting. In fact, gut microbial profiles in NAFLD without diabetes are not well characterized and remain to be explored. Moreover, data on gut microbiota alteration associated with lean NAFLD in non-diabetic individuals are still limited.

This study aimed to investigate gut dysbiosis in non-diabetic patients with NAFLD in relation to the spectrum of BMI, including lean, overweight, and obese subgroups. In addition, we attempted to identify specific microbial patterns that could discriminate among these subgroups of NAFLD. The identification of gut microbial composition according to BMI status might lead to an improved understanding of the role of gut microbiota related to NAFLD without T2DM.

## 2. Results

### 2.1. Baseline Characteristics

A total of 100 patients with NAFLD with varying BMI, including 8 patients who were lean, 17 patients who were overweight, 41 patients with obese I, and 34 patients with obese II, as well as 16 healthy subjects, were recruited in this study. There was no difference in gender, AST, ALT, total cholesterol, or triglycerides, while age, BMI, HbA1c, liver stiffness, and CAP score were significantly different among the groups (Table 1).

### 2.2. The Alpha Diversities of Gut Microbiota

After quality filtering, denoising, read merging, and chimera removal, an average of 43,961.59 ASVs per sample were obtained (Appendix A). The sequencing depths were examined by plotting the rarefaction curve from the number of species. The plot was visualized at a 25,000-read depth. Our results show that all samples reached plateaus with an average of 6275 reads (Appendix A).

To identify the alteration of alpha diversities, the Chao1, observed feature, and Shannon index among subgroups of NAFLD stratified by BMI spectra were compared to healthy controls, as shown in Figure 1. There were significant differences in those indexes between the patient subgroups and healthy controls, as all levels of alpha diversities were highest in healthy controls. However, the alpha diversity differences between the patient subgroups were not significant, except for obese I versus obese II in Chao1 and the observed feature (*p*-value < 0.05) and lean versus obese II in the Shannon index (*p*-value < 0.05). These findings suggested that community richness and diversity were not associated with the ranges of BMI.

### 2.3. The Beta Diversities of Gut Microbiota

To investigate the similarities and differences among microbial communities, the beta diversity based on the Bray–Curtis dissimilarity was analyzed, and patients with lean NAFLD were significantly clustered and differed from healthy controls (PERMANOVA, *p*-value < 0.001, Figure 2a). However, there were no significant differences in bacterial community between the subgroups of patients with NAFLD (Figure 2b–d). These data indicate that there was no distinct clustering of the gut microbiota composition based on BMI. Further subgroup analysis is shown in Appendix A. Beta diversity based on Aitchison distance was calculated from the CLR transformed at the genus level; the data showed similar results with Bray–Curtis analysis (Appendix A).

### 2.4. Alteration of Taxonomic Level of Gut Microbiota among Groups

Next, we investigated the relative abundance at the phylum level of all samples (Figure 3a). *Firmicutes* (59.5%), *Bacteroidota* (28.1%), *Actinobacteriota* (6.5%), *Proteobacteria* (3.6%), and *Fusobacteriota* (1.7%) had a high abundance of bacteria. Figure 3a did not show statistically significant differences. To investigate specific taxa, we examined the ratio of *Firmicutes/Bacteroidota* (F/B ratio), as shown in Figure 3b. Compared with the healthy controls, the F/B ratio of patients with lean, obese I, and obese II had remarkable differences, respectively (FDR *p*-value = 0.007, 0.007, and 0.002). There was no difference in the F/B ratio between healthy controls and patients who were overweight. Among the subgroups of NAFLD, the F/B ratio in patients with overweight, obese I, and obese II had no significant differences compared to the lean subgroup. Moreover, patients with obese I and obese II had no significant difference in this ratio (FDR *p*-value = 0.313). These results suggest there was no trend in the elevated F/B ratio according to BMI.

At the family level, *Lachnospiraceae* (35.1%), *Bacteroidaceae* (16.2%), *Prevotellaceae* (9.7%), *Ruminococcaceae* (9.5%), *Selenomonadaceae* (3.9%), *Coriobacteriaceae* (3.2%), *Bifidobacteriaceae* (3.0%), *Acidaminococcaceae* (2.1%), *Enterobacteriaceae* (1.9%), *Oscillospiraceae* (1.8%), *Fusobacteriaceae* (1.7%), *Veillonellaceae* (1.5%), and *Sutterellaceae* (1.3%) had major bacterial abundance (Appendix A). At the genus level, *Faecalibacterium* (9.4%), *Megamonas* (8.5%), *Blautia* (7.4%), *Fusicatenibacter* (6.8%), and *Escherichia-Shigella* (6.3%) were the major bacterial genera with high abundance (Appendix A).

To specifically evaluate significant changes in gut microbiota composition at the genus level, differential abundance analysis was performed using DESeq2. These differences in bacteria taxa between subgroups of patients with NAFLD are shown in Figure 4. Subgroup analysis for the healthy controls, lean, and non-lean patients is available in Appendix A.

To investigate the signature of differential bacteria among groups, combining DESeq2 with classical statistical tests like the Kruskal–Wallis rank-sum test with FDR correction was performed [8]. Bacteria that showed significant differences between at least one pair of these groups were chosen. Compared to the non-lean subgroups (overweight, obese I, and obese II) and healthy controls, lean subjects were remarkably enriched in *Bacteroides* (*Bacteroidaceae* family), *Megasphaera* (*Veillonellaceae* family), and *Escherichia-Shigella* (*Enterobacteriaceae* family) (Figure 5a–c), while *Coprococcus* (*Lachnospiraceae* family), *Fusicatenibacter*, *Lachnospira* (*Lachnospiraceae* family), and *Subdoligranulum* were significantly depleted (Figure 6a–d). However, only *Escherichia-Shigella*, *Lachnospira*, and *Subdoligranulum* in the lean subgroup displayed a significant difference from those of the non-lean subgroups (FDR adjusted *p*-value < 0.05.).

### 2.5. Plasma I-FABP and LBP Levels in Patients with NAFLD across the BMI Spectrum

To investigate the levels of biomarkers for gut epithelial permeability and bacterial translocation, the intestinal fatty acid-binding protein (I-FABP) and lipopolysaccharide-binding protein (LBP) levels were measured using ELISA kits, respectively. Our results demonstrated significantly lower I-FABP levels in healthy subjects than lean NAFLD patients (*p* = 0.002), as well as between healthy individuals and non-lean NAFLD patients (*p* = 0.004, Figure 7a). Additionally, there was a marginally significant increase in I-FABP levels in lean patients when compared to non-lean individuals (*p* = 0.054, Figure 7a). In subgroup analysis, Figure 7b shows significant differences in I-FABP levels between the healthy controls and the lean (*p* = 0.005) and obese I (*p* = 0.027) subgroups, but there was no significant difference in the biomarker level between healthy controls and the other subgroups of NAFLD. Notably, the lean subgroup exhibited the highest level of I-FABP, suggesting a marked increase in I-FABP in lean patients compared to their non-lean counterparts. For LBP, there was no statistically significant difference in LBP levels across the lean and non-lean subgroups (Figure 7c,d, *p* > 0.05).

Moreover, I-FABP did not show a significant correlation with any clinical parameters or gut microbiome compositions. Conversely, LBP demonstrated a weak correlation with fasting blood sugar, BMI, and CAP scores, with the coefficients of r = 0.246 (*p* = 0.029), r = 0.233 (*p* = 0.029), and r = 0.255 (*p* = 0.020), respectively.

### 2.6. Performance of Discriminating Lean from Non-Lean Using Significant Bacteria

To evaluate the diagnostic performance of significant bacterial genera in differentiating patients with lean and non-lean NAFLD, ROC curve analysis based on a logistic regression model was performed. This method has been proven as an effective ‘classification’ tool for biomedical studies [9]. In Figure 8, The AUCs of *Escherichia-Shigella*, *Lachnospira*, and *Subdoligranulum* were 0.713, 0.753, and 0.750, respectively. The combination of *Lachnospira* and *Subdoligranulum* achieved an AUC of 0.807. Interestingly, the combined AUC of *Escherichia-Shigella, Lachnospira*, and *Subdoligranulum* reached 0.821, suggesting that gut microbial composition at the genus level could effectively discriminate lean from non-lean NAFLD.

### 2.7. Predicted Functional Pathways Based on Differential Abundances

To predict the most represented functions of the microbial communities between lean and non-lean NAFLD, PICRUST2 was performed (Figure 9). There were 17 differentially abundant pathways between the two subgroups. Based on the MetaCyc Metabolic Pathway Database, creatinine degradation II, the superpathway of L-isoleucine biosynthesis I, the superpathway of adenosine nucleotides de novo biosynthesis I, the superpathway of CDP-glucose-derived O-antigen building, the superpathway of adenosine nucleotides de novo biosynthesis II, L-isoleucine biosynthesis I (from threonine), L-valine biosynthesis, L-isoleucine biosynthesis III, the superpathway of branched amino acid biosynthesis, 5-aminoimidazle ribonucleotide biosynthesis I, heparin degradation, gluconeogenesis I, the superpathway of demethylmenaquinol-6 biosynthesis II, NAD biosynthesis II (from tryptophan L-tryptophan degradation to 2-amino-3-carboxymuconate semialdehyde, 5-aminoimidazle ribonucleotide biosynthesis II, and the superpathway of 5-aminoimidazle ribonucleotide biosynthesis showed higher abundance in non-lean subjects. The functional pathways associated with ranges of BMI in subgroup analysis are shown in Appendix A.

## 3. Discussion

Gut dysbiosis, defined as the imbalance between beneficial and pathogenic bacteria usually occurs in patients with chronic liver diseases, including NAFLD, particularly those with coexisting metabolic syndromes such as obesity and T2DM [6]. However, data regarding gut microbiota characteristics in patients with NAFLD without T2DM are restricted. One of the challenges in studying the connections between gut microbiota and NAFLD is the strong effect of T2DM on gut dysbiosis, and thus, the confounding factors related to diabetes could not be excluded. Moreover, anti-diabetic drugs such as metformin might alter gut microbiota composition and interfere with the data analysis [10]. In this context, our study is one of the first reports to demonstrate the signature of gut bacteria in non-diabetic NAFLD individuals based on the full spectrum of BMI according to the Asian-BMI classification. Specifically, the patients were divided into four subgroups, including lean, overweight, obese I, and obese II. Although BMI might be considered only an estimated measurement of overweight and obesity, it provides one of the best indicators at the population level for both sexes and all age groups in adults.

In this report, we observed significant changes in gut microbiota diversity and the composition between healthy subjects and patients with NAFLD, irrespective of their BMI. In agreement with most reports, alpha diversity was significantly reduced in all NAFLD subgroups compared to healthy controls, as indicated by the Chao1, observed feature, and Shannon index. In contrast, the overall alpha diversity did not differ between the subgroups of NAFLD, indicating that community richness or evenness might not be related to patients’ BMI. This observation is in line with previous studies from Japan and Korea, demonstrating that there is comparable bacterial diversity between obese and non-obese NAFLD [11,12]. Moreover, there was no significant difference in beta diversity among the subgroups of NAFLD, indicating the comparable clustering of gut microbiota composition based on BMI.

At the phylum level, our data also showed that there was no significant difference in the gut community among subgroups of NAFLD. We then further explored the differences in the F/B ratio among the studied groups. Indeed, *Firmicutes* and *Bacteroidetes* are two dominant bacteria, accounting for over 90% of the whole microbiota in the intestine, and thus, the F/B ratio is considered to play a crucial role in modulating gut homeostasis [13]. Several cross-sectional studies in humans have demonstrated a trend in which gut microbiota differs from that of healthy subjects at the phylum level, with an increased F/B ratio in NAFLD [14]. Additionally, a systematic review indicates a rise in the F/B ratio as a biomarker of susceptibility to obesity [15]. However, accumulated evidence also indicates discrepancies in many studies where there were no such differences or even opposite results for the F/B ratio between non-obese and obese individuals [16]. In this study, our data did not identify a clear association between the F/B ratio and obesity status, as lean and obese NAFLD had comparable ratios. Thus, shifts at the phylum level in terms of the F/B ratio might not be able to entirely capture gut dysbiosis associated with obesity in patients with NAFLD, and the identification of bacterial profiles at lower taxonomic levels, such as the genus and species remains necessary.

At the genus level, patients with NAFLD showed enhanced *Bacteroides, Megasphaera,* and *Escherichia-Shigella* and the depletion of *Coprococcus*, *Fusicatenibacter, Lachnospira,* and *Subdoligranulum*, when compared with healthy controls. Our findings were, in part, similar to a recent meta-analysis, which demonstrates increased *Escherichia, Prevotella,* and *Streptococcus*, as well as declining *Coprococcus*, *Faecalibacterium*, and *Ruminococcus* in patients with NAFLD regardless of their T2DM status [17]. Among the NAFLD, we also revealed an altered microbial composition differentiating among subgroups of patients according to BMI, particularly between the lean and other subgroups. Specifically, the lean phenotype was accompanied by an enriched *Escherichia-Shigella*, while *Lachnospira* and *Subdoligranulum* were significantly diminished compared to the others. Indeed, the alteration in the abundance of these genera was previously documented in Asian individuals with non-obese NAFLD. For instance, *Lachnospira* was shown to be significantly depleted in the non-obese NAFLD group, as demonstrated in reports from Korea and China [18,19]. In addition, *Escherichia* was enriched, while *Subdoligranulum* significantly declined in Japanese patients who were non-obese compared to obese NAFLD [11]. We further combined *Escherichia-Shigella*, *Lachnospira*, and *Subdoligranulum* as gut microbial signatures to differentiate between lean and non-lean NAFLD. Our data demonstrated that this signature was adequate to distinguish lean NAFLD from non-lean individuals with high diagnostic accuracy (AUC of 0.82).

Potential pathways from gut microbiota towards the development of chronic liver disease are mainly communicated through the gut–liver axis [20]. As previously mentioned, one of the remarkable observations in our study was the distinct microbial composition in lean NAFLD compared with non-lean NAFLD and healthy individuals. Specifically, a pro-inflammatory strain, *Escherichia-Shigella*, was shown to be elevated in patients with lean NAFLD. This genus is capable of ethanol and endotoxin synthesis, leading to the generation of oxidative stress, increased intestinal permeability, and, eventually, liver inflammation. *Escherichia-Shigella* was also regarded as an important risk factor involving the progression of NAFLD to NASH in human studies [21,22]. Additionally, *Lachnospira* and *Subdoligranulum* are short-chain fatty acid (SCFA)-producing bacteria and were found to be significantly diminished in the lean NAFLD group. In line with our report, recent data also revealed a significant decrease in the abundance of *Eubacterium*, an obligately anaerobic and SCFAs-producing bacterium, in non-obese NAFLD compared to their obese counterparts [11]. SCFAs are considered to be essential in various physiological roles, including the promotion of intestinal epithelium proliferation and integrity, as well as modulating immune activity and inflammatory response [23]. Thus, a decrease in SCFA-producing bacteria might result in a deteriorated gut barrier integrity and increased intestinal permeability, which, in turn, could promote development and disease progression in NAFLD. Moreover, the bacteria are also capable of performing the dehydroxylation and deconjugation of bile acid, which have been shown to play a crucial role in the pathogenesis of NAFLD, including cirrhosis and HCC [24].

The disruption of the intestinal barrier and the subsequent translocation of bacteria or bacterial components, particularly lipopolysaccharides (LPSs), are now recognized as principal contributors to low-grade inflammation typically associated with obesity and NAFLD [25] and the development of NAFLD [26]. Currently, the LPS-binding protein (LBP) is considered an accurate biomarker of LPS and endotoxemia [27]. In our study, we observed a trend indicating an elevation in LBP levels corresponding with an increase in BMI, although this did not achieve statistical significance. We also measured the intestinal fatty acid-binding protein (I-FABP), which plays an important role in the intracellular transport and metabolism of fatty acids within enterocytes and is released into the intestinal lumen upon cellular damage [28,29]. I-FABP is linked with intestinal dysfunction and has been identified as a biomarker indicative of increased intestinal permeability, commonly referred to as a ‘leaky gut’, in various intestinal disorders such as necrotizing enterocolitis and inflammatory bowel disease [30,31]. Indeed, there is a correlation between the expression of I-FABP and metabolic changes induced by a high-fat diet in the mouse model [32]. A previous report observed that patients with chronic viral hepatitis exhibited elevated plasma levels of FABP-2 when compared to the control groups [33]. However, data regarding intestinal barrier dysfunction and I-FABP in patients with NAFLD, particularly in lean subjects, are currently limited. In this study, the increased expression of I-FABP was observed in patients who were lean and non-lean compared to healthy controls. Interestingly, patients with a lean phenotype displayed the highest level of circulating I-FABP, suggesting that lean individuals are more likely to have intestinal barrier injury than patients in the non-lean subgroups.

The “lean” phenotype is currently recognized as a district entity of the NAFLD spectrum, probably more than just an early phase or less severe manifestation [34]. However, it is still unclear whether lean NAFLD differs from obese individuals with NAFLD regarding metabolic dysfunction and long-term liver-related outcomes. A recent meta-analysis indicated that the severity of metabolic disorders is weight-dependent, and thus, lean persons exhibit less metabolic dysfunction compared with overweight/obese NAFLD [35]. In agreement with these data, our findings suggested that the microbe plays a predictive role in metabolic pathways such as L-isoleucine and L-valine biosynthesis, which were significantly enriched in non-lean compared with lean individuals. The biosynthesis of these branched-chain amino acids (BCAAs) has been associated with the progression of obesity-related metabolic disorders, including T2DM and NAFLD [36]. Moreover, gluconeogenesis was also significantly different between the lean and non-lean subgroups. Several previous studies indicate that abnormal glucose metabolism is common in NAFLD due to increased gluconeogenesis, disrupted insulin sensitivity, and consequent metabolic disorders [37].

Regarding long-term clinical outcomes, a meta-analysis revealed that individuals with lean NAFLD have an increased risk of liver-related complications compared to non-lean NAFLD [3]. Currently, the underlying mechanisms related to more advanced liver disease in lean NAFLD are not completely known. Emerging data suggest that genetic predispositions, such as single nucleotide polymorphisms (SNPs), may contribute to the manifestation of disease progression in lean NAFLD [38]. Additionally, recent data emphasize the link between an altered gut microbial composition and fibrosis severity, particularly in non-obese NAFLD compared with obese individuals [18]. Together, it seems that genetic predispositions and gut dysbiosis are strong driving forces in the development and progression of lean NAFLD. Together, these data highlight the differences between lean and overweight/obese NAFLD in terms of clinical outcome, metabolic dysfunction, genetic risk factors, and altered gut microbial profiles.

We acknowledged some limitations in our report. First, this is a cross-sectional study performed in a single center; thus, the generalization of our findings to other populations with different geographical settings is unknown. Second, the sample size was relatively small, which might lead to inadequate statistical power as a type II error. Finally, this association might not be able to propose causality, and further studies are needed to assess whether these microbial profiles and metabolites, such as SCFA concentrations, impact the pathogenesis of NAFLD. Despite these limitations, the strength of our study comprised only non-diabetic patients, and thus, the identification of microbial profiles may provide a better understanding of gut microbiota patterns specific to NAFLD without diabetes. However, additional studies with an increased number of patients are required, which would be valuable to explore the pathophysiology and potential therapy of gut microbiota, particularly in lean NAFLD.

## 4. Materials and Methods

### 4.1. Human Subjects

This report is a cross-sectional study comprising non-diabetic patients with NAFLD who were enrolled at King Chulalongkorn Hospital, Bangkok, Thailand. Inclusion criteria were the following: adult Thai patients diagnosed with hepatic steatosis via a controlled attenuation parameter (CAP, >248 dB/m) using the M or XL probes as appropriate [39]. Participants who met any of the following conditions were excluded from the study: (i) significant alcohol consumption (≥30 g for men and ≥20 g for women), (ii) the presence of other liver diseases, such as chronic viral hepatitis and autoimmune hepatitis, (iii) a history of hepatocellular carcinoma (HCC), or other types of cancers. Participants were advised against taking any antibiotics, prebiotics, or probiotic supplements for at least 3 months prior to enrollment.

For all participants, anthropometric variables, including weight, height, and BMI, were recorded. According to the Asian-BMI classification, patients with a BMI of 18.5–22.9, 23–24.9, 25–29.9, and ≥30 kg/m^2^ were categorized as lean, overweight, obese class I, and obese class II, respectively [40]. The control group consisted of healthy, lean volunteers without any liver or systemic diseases. This study adhered to The Declaration of Helsinki regarding the involvement of human subjects and obtained approval from the Institutional Review Board of the Faculty of Medicine, Chulalongkorn University (IRB no. 957/64). All participants provided informed consent for this study.

### 4.2. Fecal Collection and DNA Extraction

All participants provided fecal samples in DNA/RNA Shield™-Fecal Collection tubes (Zymo Research Corp, Irvine, CA, USA), which were stored at −80 °C until further analysis. The fecal samples were extracted for DNA using the ZymoBIOMICS™ DNA Miniprep Kit (Zymo Research Corp, Irvine, CA, USA) according to the manufacturer’s instructions. The quality of DNA and its concentration were identified using a DeNovix™ UV-Vis spectrophotometer (DeNovix Inc, Wilmington, DE, USA).

### 4.3. Microbiome Data Analysis

The FASTQ format was used for 250 bp paired-end sequences. These sequences were processed using QIIME2-2022.2 [30], where adapters and primers were removed from the start of both the forward and reverse reads through the use of q2-cutadapt. The sequences were then shortened at specific forward and reverse points. The FASTQ format was used for 250 bp paired-end sequences. These sequences were processed using QIIME2-2022.2 [41]. Subsequently, paired-end reads were combined, covering overlapping areas to represent V4 16S rRNA sequences while discarding chimeric sequences with the aid of q2-dada2. These pre-processed sequences underwent clustering into distinct sequence features following the amplicon sequence variant (ASV) approach, leading to the formation of a feature count abundance table for the samples [42]. Additionally, these features were taxonomically classified using the SILVA v138 database [43]. Rarefying normalization was employed to standardize all samples by subsampling them to an equal number of reads. The sequencing depths were examined by plotting the rarefaction curve from the number of species (Appendix A). Subsequently, taxonomic abundances were determined for all samples using the q2-classify-sklearn tool. The differential bacterial abundances between groups were analyzed and compared using the DESeq2 method [44].

### 4.4. Diversity Analysis

Taxonomic abundances at the genus level were used to calculate diversity metrics. Alpha diversity based on Chao1, the observed feature, and the Shannon index were estimated for all samples using vegan v2.6.4 [45]. Alpha diversity was compared between NAFLD subgroups using the Kruskal–Wallis test. Beta diversity in terms of Bray–Curtis dissimilarity and Aitchison distance were calculated for all samples using the permutational multivariate analysis of variance (PERMANOVA) and the permutation test for adonis under a reduced model, respectively. PCoA plots were constructed using the MicrobiomeAnalyst web base (https://www.microbiomeanalyst.ca/ accessed on 24 July 2023) [46] and R studio v4.1.0.

### 4.5. Intestinal Barrier Dysfunction and Microbial Translocation Biomarkers Measurement

To quantify the plasma lipopolysaccharide-binding protein (LBP) and intestinal fatty acid-binding protein (I-FABP), plasma was extracted from peripheral blood samples and stored at −80 °C for further analysis. LBP and I-FABP levels were evaluated using commercially available enzyme-linked immunosorbent assay (ELISA) kits (Hycult Biotech enzyme-linked immunosorbent assay kit, Uden, The Netherlands), adhering to the instructions provided by the manufacturer. The plasma samples were diluted at a proportion of 1:1000 for LBP and 1:2 for I-FABP prior to the measurement.

### 4.6. Statistical Analysis

Statistical analyses for the characteristics of participants were performed using SPSS (version 22.0.0, SPSS Inc., Chicago, IL, USA) and GraphPad Prism Software (version 9.5.0, Boston, MA, USA). Chi-square and one-way ANOVA were used to analyze category data. The nonparametric Mann–Whitney test for the comparison of unpaired groups was performed. Kruskal–Wallis was used to analyze for multiple testing with Benjamini–Hochberg correction to control the False Discovery Rate (FDR). The comparison of specific relative abundance of bacterial genera between groups was tested using the Kruskal–Wallis test with FDR correction. Spearman’s rank correlation coefficient was performed for correlation analysis. The diagnostic performance of bacterial genera was calculated using the area under the receiver operating characteristic (ROC) curve (AUC) based on a logistic regression model using SPSS [9]. An FDR-adjusted *p*-value < 0.05 or *p*-value < 0.05 was considered statistically significant.

## Figures and Tables

**Figure 1 ijms-25-01807-f001:**
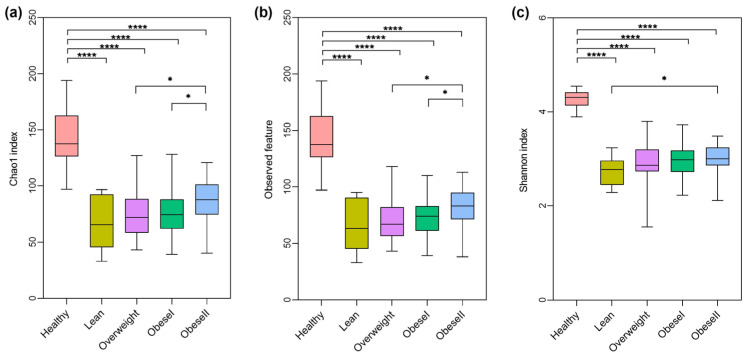
Alpha-diversity differences among healthy controls and NAFLD patients with varying BMI. (**a**) Chao1 index, (**b**) observed feature, (**c**) Shannon index, * *p* < 0.05, **** *p* < 0.0001.

**Figure 2 ijms-25-01807-f002:**
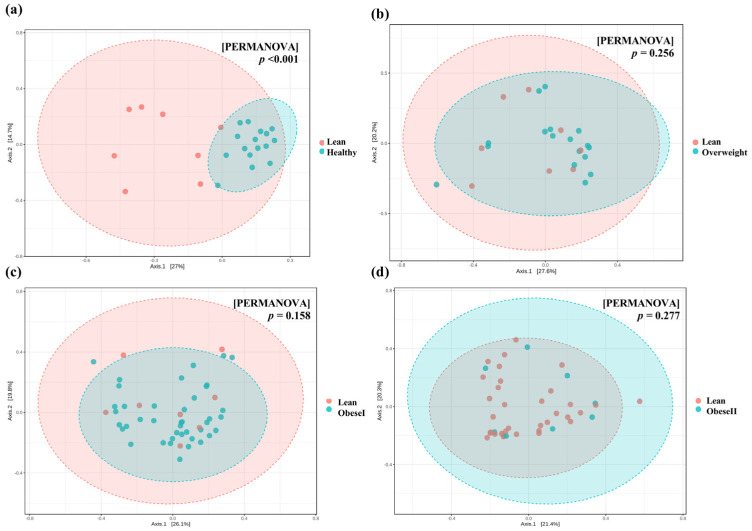
Principal Coordinate Analysis (PCoA) plots of beta diversity based on Bray−Curtis distance. (**a**) Lean vs. healthy controls. (**b**) lean vs. overweight, (**c**) lean vs. obese I, and (**d**) lean vs. obese II. Statistical analysis was calculated using pairwise PERMANOVA, and a *p*-value < 0.01 was considered statistically significant.

**Figure 3 ijms-25-01807-f003:**
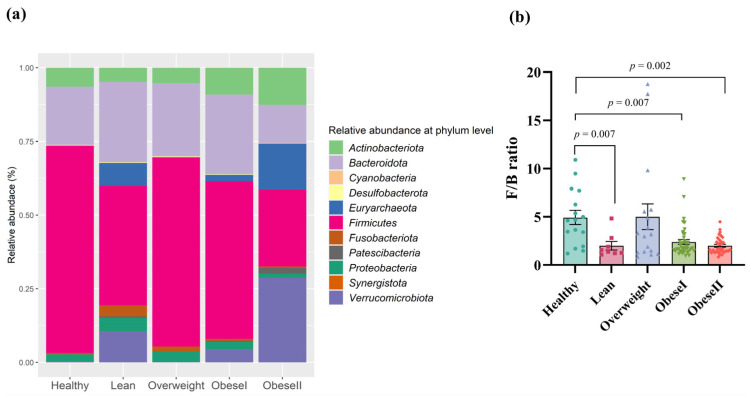
Gut microbial composition at the phylum level (**a**) and F/B ratio in patients with different ranges of BMI; (**b**) FDR adjusted *p*-value < 0.05.

**Figure 4 ijms-25-01807-f004:**
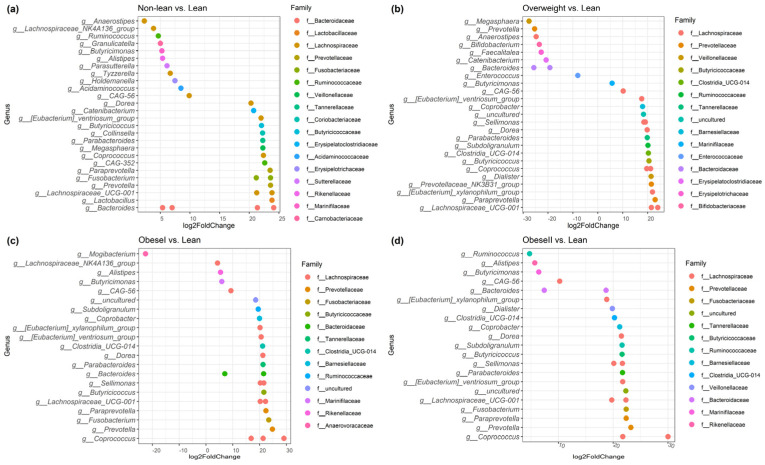
Identifying differentially abundant microbes between non−lean vs. lean (**a**), overweight vs. lean (**b**), obese I vs. lean (**c**), and obese II vs. lean individuals (**d**) using DESeq2 analysis.

**Figure 5 ijms-25-01807-f005:**
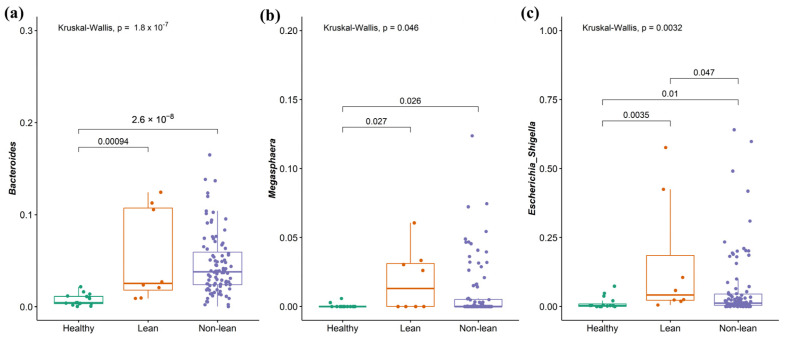
Signature of bacterial taxa enrichment in patients with lean NAFLD. (**a**) *Bacteroides*, (**b**) *Megasphaera*, and (**c**) *Escherichia-Shigella*.

**Figure 6 ijms-25-01807-f006:**
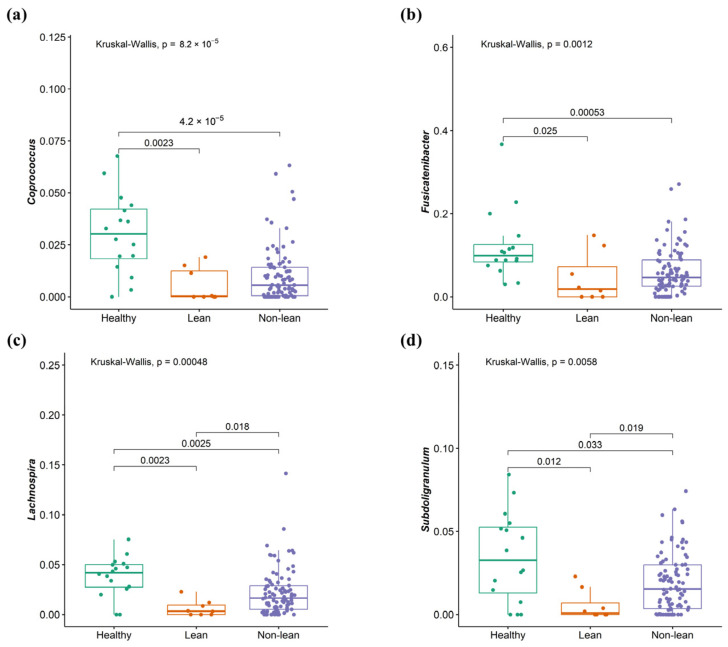
Signature of bacterial taxa depletion in patients with lean NAFLD. (**a**) *Coprococcus*, (**b**) *Fusicatenibacter*, (**c**) *Lachnospira*, and (**d**) *Subdoligranulum*.

**Figure 7 ijms-25-01807-f007:**
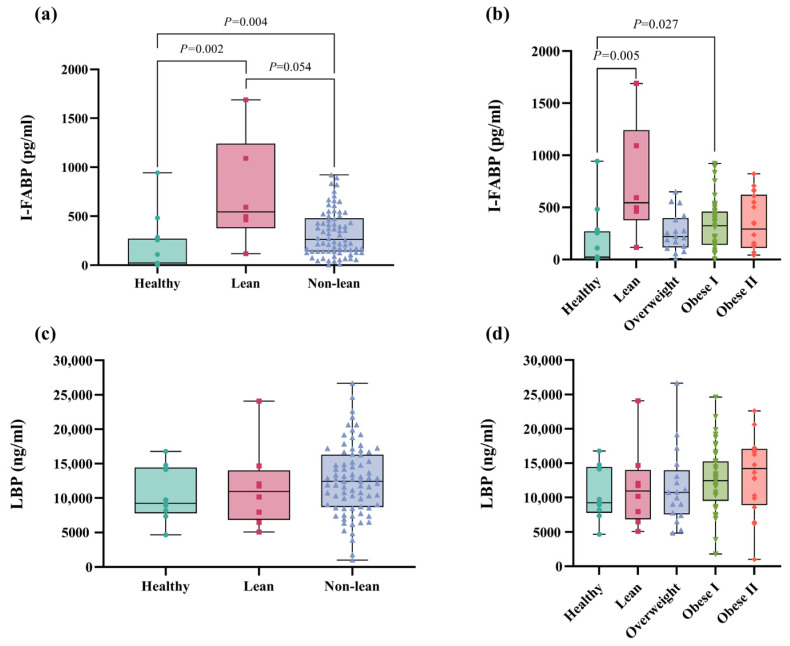
Plasma I-FABP and LBP levels in patients with lean and non-lean subgroups. (**a**) I-FABP level in patients with lean vs. non-lean. (**b**) I-FABP level in patients with NAFLD across the BMI spectrum (**c**) LBP level in patients who were lean vs. non-lean (**d**) LBP level in patients with NAFLD across the BMI spectrum.

**Figure 8 ijms-25-01807-f008:**
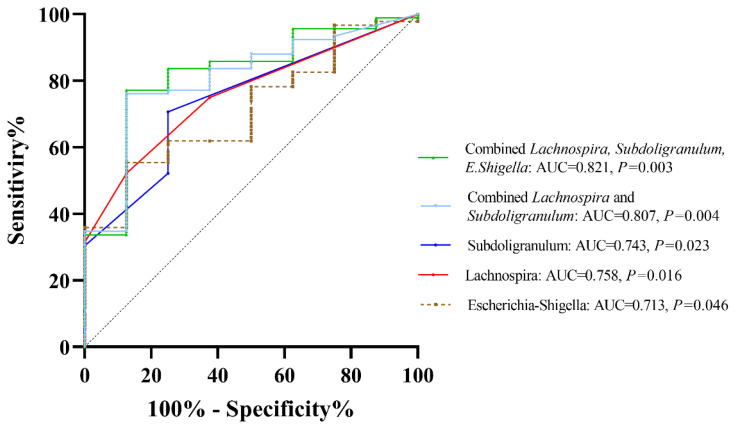
ROC curve analysis based on significant bacteria at the genus level.

**Figure 9 ijms-25-01807-f009:**
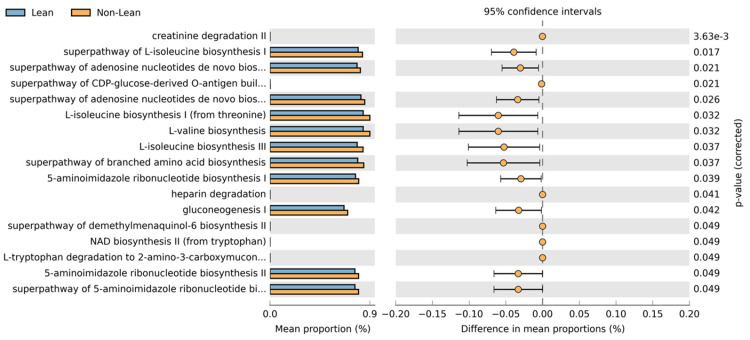
Predicted metabolic pathways using PICRUSt2 analysis based on the MetaCyc Metabolic Pathway Database.

**Table 1 ijms-25-01807-t001:** Baseline characteristics of patients and healthy individuals.

Characteristics	Healthy Controls (n = 16)	Patients with NAFLD				*p*-Value
		Lean (n = 8)	Overweight (n = 17)	Obese I (n = 41)	Obese II (n = 34)	
Gender, male (%)	8 (50)	4 (50)	12 (70.6)	24 (58.5)	16 (47.1)	0.560
Age (years)	32.69 ± 8.61	52.25 ± 12.09 ^$^	52.82 ± 16.39 ^#^	53.83 ± 11.44 ^#^	37.64 ± 8.67 ^#^	<0.001 ^#^, 0.016 ^$^
Body mass index (kg/m^2^)	21.83 ± 2.75	22.14 ± 0.88	24.21 ± 0.55	27.00 ± 1.21	37.64 ± 8.67	<0.001 ^@,฿,&,!^
						0.020 ^†^
AST (IU/L)	18.44 ± 4.64	18.40 ± 3.78	24.56 ± 6.63	28.48 ± 13.25	25.94 ± 10.23	0.058
ALT (IU/L)	19.78 ± 11.05	27.8 ± 28.93	37.38 ± 23.17	40.55 ± 28.52	36.64 ± 20.90	0.207
Total cholesterol (mg/dL)	220.25 ± 41.98	191.40 ± 20.76	205.00 ± 41.20	196.95 ± 38.19	192.72 ± 42.40	0.461
Triglycerides (mg/dL)	103.25 ± 63.50	113.40 ± 46.31	129.75 ± 46.45	127.05 ± 46.08	140.45 ± 50.98	0.372
Fasting blood sugar (mg/dL)	101.63 ± 10.13	104.83 ± 9.33	100.73 ± 15.58	106.32 ± 12.21	105.42 ± 15.50	0.681
HbA1c (%)	5.25 ± 0.16	5.45 ± 0.31	5.28 ± 0.57	5.57 ± 0.29	5.66 ± 0.44	0.034 ^฿^
Liver stiffness (kPa)	4.60 ± 0.52	9.94 ± 7.58	5.73 ± 2.20	6.14 ± 3.33	8.77 ± 5.76	0.006 ^△,▽^
Controlled attenuation parameter (dB/m)	239.00 ± 73.08	272.57 ± 45.89	312.06 ± 39.65	303.49 ± 30.42	328.55 ± 47.42	<0.022 ^@^, 0.001 ^†,!^, 0.002 ^⧫^

Data as shown in mean ± SD; n (%), median ± IQR; Different symbols indicate statistically significant differences (*p* < 0.05); ^#^ = healthy controls vs. overweight or obese I or obese II; ^$^ = healthy controls vs. lean; ^@^ = obese II vs. lean; ^฿^ = obese II vs. overweight; ^&^ = obese I vs. obese II; ^!^ = healthy controls vs. obese II; ^△^ = lean vs. overweight; ^▽^ = lean vs. obese I; ^†^ = healthy controls vs. overweight; ^⧫^ = healthy controls vs. obese I; Aspartate aminotransferase: AST; Alanine aminotransferase: ALT; Hemoglobin A1c: HbA1c. Chi-square test for category data and one-way ANOVA and post hoc with Bonferroni multiple comparisons for independent variables were performed.

## Data Availability

The datasets presented in this study were uploaded to the Short Read Archive of the National Center for Biotechnology Information (NCBI) with the Bioproject accession number PRJNA1023072 and are available on request from the corresponding author.

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
