# Peer review of "Gut Microbiota in Patients with Non-Alcoholic Fatty Liver Disease without Type 2 Diabetes: Stratified by Body Mass Index"

_ijms, 2024, doi:10.3390/ijms25031807_

Round 1

Reviewer 1 Report

Comments and Suggestions for Authors

Chuaypen N. and co-authors have reported that non-diabetic patients with lean NAFLD had a significant difference in bacterial composition compared with non-lean individuals. From their results, the authors conclude that gut microbiota signatures could be used as potential targeting therapy in patients with lean NAFLD. Upon reviewing the paper, I find areas that require attention.

1.      The main reservation is the lack of novelty of the study. In the current study, the authors do not present data regarding the severity of NAFLD or fibrosis. There are no data of SCFA concentration, gut epithelial permeability or correlations between these parameters and gut microbiota composition. Therefore, although the general concept that modulation of the gut microbiome might provide a treatment target in non-diabetic patients with lean NAFLD is attractive, the authors have not provided a mechanistic insight. Therefore, the study is highly speculative and some conclusions are not supported by the results.

2.      Microbiome analysis should be further explained. Please, indicate the sequencer used, how many sequences were obtained from samples, the rarefaction read counts, etc.

3.      Statistics section is insufficient. It is not clear when the authors have controlled for multiple testing.

4.      In figures, please, show only differences statistically significant because it is hard to read with all comparisons

Comments on the Quality of English Language

Minor editing of English language required

Reviewer 2 Report

Comments and Suggestions for Authors

A. brief summary

The manuscript on hand is an observational study presenting the gut microbiota’s signature in non-diabetic individual with NAFLD, especially focusing on lean NAFLD individuals. Authors have successfully tested the hypothesis that a distinct microbiome signature associated with NAFLD, focusing on lean individuals. Further, the authors discovered that there is a decrease in bacterial diversity in these individuals compared to healthy individuals. 

Both, diversity of the microbiome and microbiome signature is discussed with respect to BMI only. The manuscript is relevant for the field because NAFLD is high prevalent worldwide, especially in western population, therefore understanding it’s mechanisms underlying disease, including those associated with the gut microbiome, is important in the context of therapeutic approaches. It is presented in well-structured manner however strong concerns arise while reading the manuscript. 

B. General concept comments

The methods used for overall microbiome analysis are outdated and information is missing in many cases. Therefore, the reproducible of the results based on the details given in method section is questionable. Cut-off for rarefication step is not mentioned. Detailed explanation on choosing this cutoff should be provided. For diversity analysis, it is suggested that authors should also calculate aitchison distance as it can avoid compositional bias. Methods like Deseq2 have a high false discovery rate and are not suitable for microbiome analysis. Therefore, the results are not reliable. It is suggested that the authors should use methods such as ANCOM-BC or ALDEX2. These methods account for the compositional nature of microbiome unlike Deseq2. Information regarding statistical analysis for pathways is completely missing. Therefore the conclusions are inconsistent with the evidence and arguments presented. 

C. Specific Comments

 Information shown in the taxonomy bar plots (figure 3a, figure S2, figure S3) is repeated in text (line 110-122,line 127-134). It is unclear whether P-value is FDR corrected or not for figure 5 and figure 6. Please perform FDR correction for all analysis if not performed. The diagnostic model used for performing ROC AUC analysis is not clearly mentioned. 

Comments on the Quality of English Language

English editing is recommended

Round 2

Reviewer 1 Report

Comments and Suggestions for Authors

The authors have addressed my comments and the revised manuscript has improved substantially. However, still some issues exist.

Comments:

1.       An important concern arises regarding how the authors have selected the bacterial genera on which they have focused their work. It appears that the authors chose bacterial genera with differential abundance between lean and non-lean groups based on DESeq2 analysis. If this is the case, why are most of the data presented in Figures 5 and 6 not significantly different? Additionally, the data distribution in both groups is very similar, and differences seem to be related to the sample size. Escherichia/Shigella does not appear in Figure 4. Please provide clarification.

2.       Since the aim of the study is to investigate gut dysbiosis in non-diabetic patients with NAFLD in relation to the spectrum of BMI, the signature of bacterial taxa in the different groups of non-lean patients should be presented.

Minor comments:

3.       Figure 3a does not show statistically significant differences. Please modify the text (page 4, line 120).

4.       Legend of Figure 1: please correct ≤.

5.       Page 6, line 148: correct "Bacteroides."

6.       The size of Figure 4 should be changed to improve its quality.

7.       Page 12: Please check section 4.3 since some sentences have been duplicated but cited with different references.

Comments on the Quality of English Language

Minor editing of English language required

Round 3

Reviewer 1 Report

Comments and Suggestions for Authors

Thank you very much for addressing my comments. I believe that the manuscript has improved substantially.

The key issue of the manuscript was how the bacteria genera had been selected, and it is important that this information be clearly presented in the manuscript. Please include the sentence “Bacteria that showed significant differences between at least one pair of these groups were chosen” in page 6, lines 151-152